# Role of Digital Health and Artificial Intelligence in Inflammatory Bowel Disease: A Scoping Review

**DOI:** 10.3390/genes12101465

**Published:** 2021-09-22

**Authors:** Kamila Majidova, Julia Handfield, Kamran Kafi, Ryan D. Martin, Ryszard Kubinski

**Affiliations:** Phyla Technologies Inc., Montréal, QC H3C 4J9, Canada; majidova.kamila@gmail.com (K.M.); julia.handfield@phyla.ai (J.H.); kam@phyla.ai (K.K.); ryan.martin@phyla.ai (R.D.M.)

**Keywords:** inflammatory bowel disease (IBD), Crohn’s disease (CD), ulcerative colitis (UC), digital health (DH), artificial intelligence (AI), diagnosis, treatment, monitoring, prognosis

## Abstract

Inflammatory bowel diseases (IBD), subdivided into Crohn’s disease (CD) and ulcerative colitis (UC), are chronic diseases that are characterized by relapsing and remitting periods of inflammation in the gastrointestinal tract. In recent years, the amount of research surrounding digital health (DH) and artificial intelligence (AI) has increased. The purpose of this scoping review is to explore this growing field of research to summarize the role of DH and AI in the diagnosis, treatment, monitoring and prognosis of IBD. A review of 21 articles revealed the impact of both AI algorithms and DH technologies; AI algorithms can improve diagnostic accuracy, assess disease activity, and predict treatment response based on data modalities such as endoscopic imaging and genetic data. In terms of DH, patients utilizing DH platforms experienced improvements in quality of life, disease literacy, treatment adherence, and medication management. In addition, DH methods can reduce the need for in-person appointments, decreasing the use of healthcare resources without compromising the standard of care. These articles demonstrate preliminary evidence of the potential of DH and AI for improving the management of IBD. However, the majority of these studies were performed in a regulated clinical environment. Therefore, further validation of these results in a real-world environment is required to assess the efficacy of these methods in the general IBD population.

## 1. Introduction

### 1.1. What Is Inflammatory Bowel Disease?

Inflammatory bowel disease (IBD), which encompasses ulcerative colitis (UC) and Crohn’s disease (CD), is a chronic disease characterized by an uncontrolled inflammatory response in the gastrointestinal tract that commonly follows a relapsing and remitting course [1,2]. Common symptoms of IBD include abdominal pain, diarrhea, rectal bleeding, fatigue, and extraintestinal manifestations of the disease [2]. The disease and its symptoms can lead to substantial morbidity and absenteeism, a higher cost of living, and a diminished quality of life [1].

UC is characterized by episodes of inflammation limited to the mucosal layer of the colon. In the majority of cases, inflammation involves the rectum and often extends to more proximal portions of the colon in a continuous fashion [3,4]. Assessments of severity can vary based on the specific index or score used, such as the Montreal classification of IBD and the UC Colonoscopic Index of Severity [3,4,5]. The severity of UC is generally classified as mild, moderate, or severe disease, or UC in clinical remission, which is denoted when the patient has asymptomatic disease [3,4]. Practice guidelines stratify patients into either a low- or high-risk category by assessing inflammatory status in order to estimate the risk of long-term sequelae such as colectomy [3].

CD is defined by transmural inflammation and by discontinuous areas of involvement across any area of the gastrointestinal tract, such that segments of non-inflamed bowel can be interrupted by areas with signs of disease [3]. The transmural inflammatory nature of CD may lead to fibrosis, strictures, and obstructive clinical presentations that are not typically seen in patients with UC. Transmural inflammation may also result in sinus tracts, giving rise to microperforations and fistula formation. CD disease activity and severity are classified using systems such as the Montreal classification of IBD and the Lémann score [3,4,6]. The clinical classification of CD is traditionally designated by age of onset, location of disease activity, and disease behaviour [4].

#### Pathogenesis

The pathogenesis of IBD relies on complex interactions between genetics, the intestinal microbiome, and the immune activity of the host as well as external environmental factors such as diet [7,8,9,10,11]. While it is known that genetic susceptibility to the disease varies between subphenotypes of IBD, knowledge of the genetic basis for its pathophysiology continues to evolve. Genome-wide association studies (GWAS) and next-generation sequencing studies have revealed over 240 unique genetic loci that contribute either to the risk of developing IBD, the disease severity, or the IBD subphenotype in a given patient [12,13,14]. Although the majority of IBD risk loci are quite pervasive, the loci exhibit low penetrance; only a fraction of individuals who harbour susceptible gene loci will develop IBD [13,15]. In addition, identified disease-related genetic loci display varying levels of penetrance between geographic populations [15]. This genetic variation suggests that heterogeneity between geographically distinct patient populations can be attributed to divergent genetic and environmental triggers [15,16,17,18]. In order to further elucidate the underlying genetics, applications of sophisticated laboratory and computational approaches are required, such as the integration of laboratory biomarker and clinical data with gene expression profiles using artificial intelligence. 

In addition to genetic factors, distinct environmental exposures, such as dietary components and gut bacteria, have been linked to development of IBD. In Western countries, there has been a consistently high prevalence and incidence of IBD since the middle of the 20th century [15,17,18,19]. However, the incidence of IBD has only recently started to rise in newly industrialized nations across Asia, Africa and South America [16,18]. The increasing incidence correlates with the propagation of highly processed foods, refined carbohydrates, and saturated fats in newly industrialized nations [20]. The integration of these elements into one’s diet is thought to affect the composition of gut bacteria and impact intestinal health.

In conjunction with genetic susceptibility to IBD, exposure to novel environments can lead to an imbalance of gut bacteria that may trigger the onset of IBD [19]. Microbial dysbiosis is characteristic of IBD and consists of features such as decreased intestinal biodiversity, altered gut microbiota composition, abnormal spatial distribution of microbes, and abnormal interactions between strains of microbiota and the host [21]. Although it remains unknown whether intestinal dysbiosis causes IBD or occurs as a result of the pathology [22], it is clear that an interference in gut homeostasis contributes to IBD pathophysiology. Several of the IBD-associated susceptibility genes are implicated in the activity of the microbiome [23,24]. Disease-related genetic variants associated with host-microbiome functions suggest that these loci impede the body’s ability to maintain tolerance against commensal gut bacteria and thus impair immune activity [24].

Furthermore, the altered interaction between the host and microbiome is accompanied by a dysregulated response of the mucosal immune system against the microbiota that reside within the intestinal lumen. In addition, the presence of autoantibodies and peptide antibodies can trigger inflammation and extra-intestinal manifestations of the disease [25]. The aberrant immune activity can be due both adaptive and innate immune responses that cause excessive reactivity and inflammation [26]. However, the use of immunocompromising drugs such as corticosteroids and biologics may increase the risk of developing infections due to the expansion of opportunistic pathogens that lead to a dysbiotic state of the gut flora [26,27]. The dysregulation can take the form of disruptions of the epithelial layer [28,29,30], excessive immune cell recruitment and activation [31,32,33], and dysregulation of secreted mediators [34]. Future studies are required to elucidate the roles of abnormal intestinal microbiota and immune cell activity and translate these findings to expand on the current approaches to care and improve clinical outcomes of patients.

### 1.2. Current Standard of Care for Inflammatory Bowel Disease

A diagnosis of IBD is primarily confirmed through endoscopy, ideally through a colonoscopy accompanied by intubation of the terminal ileum [35]. Biopsies are collected from regions of the gastrointestinal tract from tissue that appears healthy as well as regions where disease activity is evident [2]. In addition to endoscopies, there are also several biomarkers of inflammation that can be indicative of IBD. Levels of C-reactive protein (CRP), complete blood count (CBC), and liver enzymes from the blood and fecal calprotectin (FC) from stool samples can be measured to monitor disease activity and assess treatment response [2,35]. Furthermore, imaging procedures such as computed tomography enterography, magnetic resonance enterography, and magnetic resonance imaging can also be performed to further aid in diagnosis [2].

Once diagnosed, a treatment plan is prepared for each patient. As there is no cure for IBD, the goal of treatment is to reduce symptom severity, which in turn improves quality of life and helps to prevent disease progression and complications such as bowel perforation and colorectal cancer [35,36]. Mucosal healing is an emerging goal of treatment because it has been associated with positive long-term health outcomes such as fewer hospitalizations and surgeries among patients [37]. In order to reduce inflammation and enable mucosal healing in the gut, pharmacotherapy is the traditional approach to treating IBD [35,36]. Current pharmacological agents include anti-inflammatory drugs, immunosuppressants, and biologics [35,36]. In addition to pharmacological approaches, surgery is another option for treatment of IBD [38,39]. Options include proctocolectomy, which is the excision of at least part of the colon and rectum, as well as intestinal or ileo-cecal resection, among several other interventions [40]. Both prior to and after surgery, management of IBD includes prophylactic use of biologics such as anti-TNFɑ agents in a treat-to-target approach to reduce the risk of disease recurrence [41]. 

Following a diagnosis, frequent monitoring of IBD patients is crucial for assessing disease progression, preventing complications, optimizing treatment, and evaluating risks of disease relapse [37,42]. Rather than simply focusing on treating symptoms as they arise, monitoring supports longitudinal patient care [37,42]. Currently, a gold standard for monitoring IBD does not exist; however, colonoscopy, serum and fecal biomarker tests, and clinical assessment tools such as the CD Activity Index are commonly used. CRP and FC assays are consistently employed by clinicians to evaluate the potential for patient relapses from clinical remission or following an operation. However, optimal monitoring approaches to assess the health of patients both in remission and with active disease must be updated [37,42]. 

Prognostication is an important component of disease management and complement to IBD monitoring [43]. Clinical prognosis is measured by assessing the risks of disease progression based on factors such as age at the time of diagnosis, ileal disease location, and extent of bowel involvement [44]. Establishing a prognosis enables physicians to predict the course of disease and thus the well-being of patients. However, prognosis can be highly variable between patients and the role of IBD-related genetic variants in disease prognosis has not been fully characterized. A more sophisticated understanding of the biological basis for disease prognosis could contribute to better guided therapeutic approaches [45]. 

In order to monitor treatment response, disease activity, and disease progression, regular follow-up appointments are required. However, the longitudinal care needed to support those with chronic illnesses such as IBD poses a significant burden on both patients and healthcare systems. The higher healthcare utilization leads to a large financial burden to the patient and society, with estimated costs of $1.6 billion in Canada [46] and $14.6 billion to $31.6 billion in the United States [47]. Furthermore, inadequate patient monitoring ultimately increases healthcare utilization due to undetected complications, disease activity, and unresponsiveness to treatment that require attention [35,48,49]. In addition, the frequency of appointments and the need to tend to symptoms leads to greater rates of absenteeism, which can impact education [50]. Absenteeism can also affect work productivity and opportunities [51], accounting for at least $979 million and $3.6 billion in losses each year in Canada and the United States, respectively [46,52]. Timely access to care is another prevalent issue; Canadian patients wait an average of 66 days after receiving a referral to see a specialist, yet the recommended timeline is two weeks [53]. Patients living in rural areas are especially affected by these limitations due to limited access to and distance from adequate healthcare resources. Rural patients record fewer appointments with gastroenterologists, leading to more costly uses of health services such as hospitalizations and emergency room visits [54]. Therefore, innovative approaches to providing accessible healthcare are needed [54]. Digital health (DH) and artificial intelligence (AI) have the potential to address the challenges posed by longitudinal care, societal and individual burdens of chronic illness, and rural health disparities.

### 1.3. The Roles of Digital Health and Artificial Intelligence in the Care of Inflammatory Bowel Disease

Applications of DH and AI in IBD represent two novel avenues of research to improve health outcomes of IBD patients (see Figure 1). DH is a conceptual infrastructure of different digital technologies, such as mobile health applications, wearable devices, telehealth, and telemedicine, that promote positive health outcomes through accessible, efficient delivery of healthcare services and remote monitoring of patients [55]. DH for IBD offers patients more opportunities to access care, understand their health, assess options for preventative measures, receive earlier diagnoses, manage chronic diseases, and relieve the financial burden of their illness [55]. Similarly, AI has potential to become an increasingly valuable tool in healthcare spaces. AI is able to carry out computational tasks on complex data modalities at a significantly faster pace than humans. Machine learning (ML) is a subset of AI that accomplishes this function through the development of algorithms that are trained to recognize important patterns in a dataset. ML algorithms can learn relevant features from existing patient diagnosis and outcome data, which in turn can be used to predict a new patient’s diagnosis and prognosis [56]. This has many implications for the care of IBD patients because of the disease’s multivariate and dynamic nature, complicating diagnosis and long-term health monitoring. Overall, DH and AI methods have the capacity to support patients throughout the process of disease management and improve outcomes. 

#### 1.3.1. Digital Health

Several types of DH are starting to be utilized to monitor and support chronic disease patients. For instance, mobile-based remote monitoring applications allow IBD patients to report their symptoms and in turn receive educational information and tools to manage their illness [57]. The monitoring applications have resulted in reduced burden of care, better quality of and access to care, and improved patient satisfaction [57]. Mobile applications can also help users by prompting or encouraging behavioural changes to improve their health, resulting in improved self-management and reduced healthcare utilization [58]. In addition, remote management through online web-based services such as telemedicine applications can reduce healthcare utilization as well as improve quality of life and treatment adherence of IBD patients compared to standard care with in-person or telephone appointments [48]. Overall, remote disease management and monitoring through DH improves healthcare accessibility without sacrificing quality of care. 

#### 1.3.2. Artificial Intelligence

AI is emerging as a valuable tool for IBD diagnosis and to predict disease activity, future symptom severity, or treatment response [59]. In the context of IBD, there are several types of patient biological data that can be used as input for ML models, such as gut microbiota composition and gene expression, endoscopic imaging, histologic imaging, and biomarkers of inflammation in the tissue and blood [60,61,62,63,64]. In conjunction with AI, data compiled from blood biomarker tests, genome-wide association studies (GWAS), whole-genome shotgun sequencing, exome sequencing, or 16S rRNA sequencing can help unravel the interactions between genetic, molecular, and intestinal microbiota factors that influence the development and progression of IBD [63,65,66,67]. 

As the applications of AI for IBD continue to expand, one important area of growth is the significant value provided by predicting treatment response to biologic therapies. This is crucial because approximately one-third of patients do not respond to anti-TNFα therapy, while the response to treatment among other patients may decline with extended use [68]. Although biologic agents are an increasingly common option, they are expensive, costing at least US$20,000 each year [69]. The high costs associated with biologics emphasize the importance of anticipating the potential response of a patient to this therapy. Assessing responsiveness prior to administration of a biologic agent represents a cost-effective method of individualizing treatment approaches. Recent studies have demonstrated the use of AI for predicting treatment response among IBD patients, which can circumvent unnecessary or ineffective treatments to more efficiently achieve a therapeutic target. For instance, a random forest model determined fecal microbiota signatures to predict response to the biologic ustekinumab [70]. Similarly, clinical and serological data were used to develop a random forest model that could assess baseline inflammation of CD patients in order to predict their response to infliximab [71]. This research demonstrated the potential of AI-based techniques for assessing responsiveness to a given biologic therapy, providing the basis for future personalization and cost reduction in biologic treatment.

IBD is a complex disease that requires sophisticated research and computational approaches to improve diagnosis, treatment, monitoring, and prognosis. The analysis of multimodal datasets with AI shows promise in improving the current standard of care via increased personalization across all aspects of disease management. In parallel, advancement of DH technology can lead to increased accessibility of care for chronic illness patients, eventually reducing disease burden and associated costs. This scoping review serves to demonstrate the current states of DH and AI that address challenges for the care of IBD patients.

## 2. Methods

### 2.1. Scoping Review 

This scoping review permits researchers to investigate the current literature through an exploratory lens and answer a large and complex question by gathering and assessing the literature on the topic [72]. By recognizing key themes and current gaps in the areas of DH and AI as they pertain to the care of IBD patients, we can build on current research and direct new research questions.

### 2.2. Research Question 

What is the role of digital health and artificial intelligence in the diagnosis, treatment, monitoring, and prognosis of inflammatory bowel disease? 

### 2.3. Identifying Articles in Published Literature 

A search was performed using PubMed, Ovid and CINAHL with the search terms (“IBD” OR “UC” OR “CD”) AND (“DH” OR “AI”) AND (“prognosis” OR “diagnosis” OR “monitoring” OR “treatment”) or other related search queries (See Appendix A for full list). The search terms were used to identify articles published between January 2010 and February 2021. When search results were literature review articles, the research referenced in these texts were also considered for article selection. All hits from the search were initially triaged by a single person and those selected were subsequently discussed with the other authors. 

### 2.4. Article Selection 

First, the titles and the abstracts of the articles provided were evaluated. Articles were screened and excluded if they were written in a language other than English, did not have an abstract, were not written about adult patients, or were not related to IBD and AI or DH as well as diagnosis, treatment, monitoring, or prognosis of the disease. The articles that remained after this stage were read in full and screened for a second time, at which point they were excluded if they were deemed low quality according to the Jadad scale and SRQR checklist, when applicable [73,74]. Their credibility was considered by assessing biases, such as selection, information and confounding bias, and conflicts of interest. The articles that met this set of criteria were chosen for review for this paper. 

### 2.5. Data Charting 

Each article was reviewed and classified based on which aspect of living with IBD the article focused on: diagnosis, treatment, monitoring, or prognosis. 

### 2.6. Collation and Summary 

We explored the impact of DH and AI on IBD through four main areas of the course of disease: diagnosis, treatment, monitoring, and prognosis. The purpose of this scoping review was to assess the role of DH and AI in the care of patients with IBD in order to inform physicians and specialists about the different applications of these technologies and thus optimize care. This serves to summarize the literature on the uses of DH and AI with genetic and microbial data such that future studies can expand on these topics and determine additional functions of DH and AI for improving support for individuals living with IBD. 

## 3. Results

### 3.1. Overview 

A search on PubMed, Ovid, and CINAHL produced 341 results. From these results, 50 articles were duplicates. The abstract and titles of 291 articles were screened based on inclusion and exclusion criteria and 54 articles were fully read. From these, 21 articles were selected for this review, based on the inclusion and exclusion criteria (see Figure 2). To characterize the studies, 20 articles were of quantitative style and 1 of qualitative style. There were 12 articles about DH and IBD and 9 about AI and IBD. Table 1 provides an overview of the 21 articles and explains how they relate to the course of IBD.

### 3.2. Diagnosis

#### AI in Diagnosis

Although there are many tools available to diagnose patients with IBD (see Figure 1), the process often involves misclassification of the disease and its subtypes as well as repeated examinations in order to produce a consensus surrounding the diagnosis [94]. Emerging technologies and approaches, such as AI-assisted diagnostics, are needed in order to refine and optimize the diagnosis process. Different data modalities involved in the pathophysiology and diagnosis of IBD, such as genetic data and endoscopic and biopsy images, can be used as input to an AI algorithm designed to then output a predicted disease diagnosis. 

In the case of differentiating between healthy individuals and those with IBD, ML algorithms have been applied to exome sequencing data [86]. In order to distinguish between the patient groups, the authors of this study applied non-negative matrix tri-factorization to the exomes of 56 individuals (nCD = 42, nHealthy = 14). Additional biological data, including a gene’s disease-involvement in CD, potential pathogenicity, and potential functional interactions with other genes, was integrated with the exome sequencing data. The combination of exome sequencing and biological data led to a prediction area under the curve (AUC) of 0.816 following optimization of the non-negative matrix tri-factorization method. Although this study did not conduct a model validation on a held out portion of their dataset, it demonstrated that biological annotations of genes can be useful in aiding with exome data-based classification, with the additional benefit of improving clinical interpretability of a model. The co-clustering of genes and CD-associated individuals was an important method conducted by the investigators because it enabled this interpretability and can further serve to elucidate the genetic basis of CD. The labelling of exome sequences refines understanding of the relationship between disease susceptibility and genetic factors such that personalized treatment plans can be developed. While the biological annotation of exome data can invite future opportunities for personalized clinical approaches to care, validation with larger and more heterogeneous cohorts is still required [86]. 

The use of AI in the context of IBD diagnosis can also occur at different steps of the diagnostic pathway. For example, it is often difficult to discern, especially in early disease, whether a patient is suffering from CD or UC. A study of 5128 UC patients and 875 CD patients aiming to improve the differential diagnosis between UC and CD achieved good results using text descriptions of images from colonoscopies and a random forest (RF) model for classification [87]. The descriptions were pre-processed through word segmentation, selection of keywords to isolate informative terms within texts, and non-negative matrix factorization (NMF) in order to improve interpretability of extracted features and make the results more understandable for physicians. The data processed by NMF into a feature matrix and weight matrix were applied to an RF. The RF model was able to distinguish between UC and CD samples with a sensitivity of 0.890, a specificity of 0.837, and an AUC of 0.936 [87]. This ML approach is an accurate tool to achieve endoscopic diagnosis of UC and CD that could be utilized in order to support physicians.

Another important aspect of IBD diagnosis is the assessment of disease severity. Although endoscopy and patient questionnaires are good tools for assessing disease severity, it is also important to assess the disease activity at a microscopic level to allow for a characterization of damage to epithelial cells and evaluation of presence of malignancy. With this purpose, an AI system was used to accurately assess intestinal crypt architecture distortion and mucosal damage from biopsies of IBD patients. A probabilistic neural network (PNN) was used to classify colonic biopsies as normal, mild, moderate, or severe in terms of distortion of the crypt architecture, with normal crypts being the only instance where an IBD diagnosis would not be made. A total of 130 colonic biopsies were collected, 43 of which were from healthy participants and 87 of which were from patients with variable severity of IBD. The ground truth classification to train and test the PNN was the consensus diagnosis from three experts. The PNN was trained with 79 biopsies and performance assessed on 39 biopsies. Overall, the diagnostic system based on a PNN classifier performed with 98.31% precision and recall, demonstrating a successful process to automate the quantification of crypt distortions and mucosal damage [88]. Although these results are encouraging, it is important to recognize certain biases that could lead to such results, such as dataset balancing, which can result in misleading algorithm performance metrics if not properly accounted for, and a small sample size. Therefore, AI-enabled approaches present opportunities to produce graded assessments of mucosal inflammation and damage, but again, require further testing in a relevant clinical setting and on larger datasets.

The variety of approaches to accurately and efficiently diagnose IBD demonstrates the impact of ML on IBD healthcare. RF models have performed differential diagnoses between UC and CD [87], a PNN classifier was utilized to automate the analysis of distortions of intestinal crypt architecture and mucosal damage [88], and non-negative tri-matrix factorization of exome sequencing has enabled accurate CD diagnoses [86]. These systems can be applied to further personalize healthcare and provide better care to IBD patients through efficient diagnosis.

### 3.3. Treatment

Despite medical advancements such as the introduction of targeted biologic treatments, the effect of a given medication or set of medications on each IBD patient remains a challenge to predict [95]. Due to the risk of severe side effects and lack of response to pharmacological treatments, such as anti-TNFɑ therapy, predicting response to medication is important [96]. This personalization of therapeutic approaches for each patient prevents undue detriments to their QoL, which usually occur when therapy selection is prolonged [96]. New methods for prediction of treatment effectiveness, such as AI algorithms and DH monitoring systems, serve to improve and effectively personalize therapeutic selection.

#### 3.3.1. AI in Treatment

Thiopurines are immunomodulators that are a common first-line therapy for IBD [97]. However, these agents have a narrow therapeutic window, where low doses result in insufficient treatment response and high doses can result in significant immunosuppression and toxicity [90,98,99,100]. Monitoring response to this medication requires costly thiopurine metabolite testing to assess 6-thioguanine nucleotide (6-TGN) levels, which differentiates therapy responders from non-responders, and frequent hospital visits, creating an added burden for both patients and the healthcare system [90]. To mitigate these limitations, an RF algorithm was developed to predict clinical response and non-adherence to thiopurine therapy. Non-adherence refers to patients missing at least 20% of their prescribed treatment over time [101,102]. Age and laboratory biomarker data such as neutrophil count, alkaline phosphatase levels, red blood cell distribution width, and white blood cell count were utilized to distinguish between clinical responders and non-responders to this treatment. Unlike metabolite testing, laboratory tests required for data acquisition are cost-effective and readily available. An RF model accurately predicted clinical response with an area under the receiver operator curve (AuROC) of 0.856 (95% CI, 0.793–0.919) [90]. By contrast, thiopurine metabolite blood levels alone led to an AuROC of 0.594 (95% CI, 0.546–0.642), which was significantly less accurate than the use of laboratory data (*p* < 0.001) [90]. In addition, non-adherence to treatment was also identified accurately using an RF model and the clinical laboratory values, with mean platelet volume, hemoglobin, mean corpuscular volume, protein, and mean corpuscular hemoglobin as the features most important for prediction (AuROC = 0.813; 95% CI, 0.763–0.863) [90]. Therefore, AI-based prediction of personalized thiopurine dosages could reduce the need for metabolite testing by predicting treatment response and adherence, relieving the burden on both patients and healthcare systems [90]. 

A similar approach was used in the context of vedolizumab therapy for the prediction of corticosteroid-free endoscopic remission [91]. Similar to thiopurines, the costs of biologics such as vedolizumab are high, and the response is much slower than other medications and biologic agents, creating a pressure from both patients and insurance providers for predicting responders and non-responders. In a study of 491 UC patients, an RF model was developed with clinical data such as age, weight, vedolizumab dosing interval, and standard laboratory tests, such as FC and albumin, and blood levels of vedolizumab. After 6 weeks of vedolizumab therapy, levels of FC, albumin, and vedolizumab, and slopes of FC and vedolizumab levels were highly indicative of a positive outcome at 52 weeks. The algorithm successfully predicted corticosteroid-free endoscopic remission at 52 weeks with an AuROC of 0.73 (95% CI, 0.65–0.82) [91]. While AI has strong potential for improvements in the treatment of IBD patients, DH technologies can provide support for patients during therapy as well. 

#### 3.3.2. DH in Treatment

The efficiency, accessibility, safety, and scalability of DH interventions enable enhanced healthcare delivery, which is valuable for chronic disease patients [103]. The relapsing and remitting course of IBD leads to challenges for determining a suitable, personalized treatment plan that aligns with their disease activity. Therefore, the value of DH tools to support patients throughout this process cannot be understated. Notably, DH has shown to improve treatment adherence and medication management [75,76]. 

In a trial of 333 patients with mild-to-moderate UC, the “Constant-care” web service allowed users to record symptoms and recommended intake of the medication 5-aminosalicylate (5-ASA) in the event of a user reporting an acute increase in symptoms [75]. Adherence to 4 weeks of acute treatment, knowledge of IBD, and QoL all significantly improved among Constant-care users compared to patients who received standard care in an outpatient clinic (*p* < 0.05) [75]. For the cohort of patients, the Constant-care service was beneficial for the management of symptom flare-ups. Similarly, another trial evaluated the efficacy of mesalazine therapy using the guidance of the Constant-care web service. The study followed a total of 95 patients with mild-to-moderate UC who were non-adherent to 5-ASA treatment [76]. The service provided guided therapeutic interventions by promoting, discouraging, or encouraging maintenance of their dose of mesalazine, with the goal of improving patient adherence to mesalazine and achieving remission. By week 5 of the study, dosing of mesalazine was reduced in 50% of participants following Constant-care and by the end of the 12 weeks, 88% of patients had a lower dose than what they started with. After 3 months of this web-guided therapy, 86% were adherent to this approach, of which 88% continued to medicate with mesalazine and 12% required medication switch. Medication changes occurred when patients had worsening symptoms and required rescue therapy consisting of corticosteroids, immunosuppressants or biological therapies in order to modulate their disease activity [76]. Quality of life was also measured using mental and physical (MSC and PSC) health summary measures, which both increased significantly throughout the study (*p* < 0.01). The Constant-care DH platform optimized maintenance treatment for UC, increased treatment adherence, improved disease activity, and better QoL of users [76]. 

Beyond mobile applications, the use of “virtual clinics” has also been investigated as a way to improve patient disease management and QoL. These clinics involve remote consultations and patient monitoring via telehealth approaches, improving accessibility and relieving the burden of care, especially for those in remote areas. Srinivasan et al. (2020) found that anti-TNF treatment (infliximab and adalimumab) was more successful in CD patients participating in a virtual outpatient clinic (*n* = 149). In addition, patients in the virtual clinic described it as a more positive way to receive support and coordinate care compared to those treated through standard care. The virtual clinic was able to shorten the time to treatment success (log rank test, *p* < 0.001), provide more suitable dose intensification (82.6% for virtual clinic participants vs. 40.0% for standard care patients, *p* < 0.001), and improve disease control (84.1% vs. 28.8%, *p* < 0.001) and treatment de-escalation (21.3% vs. 10.0%, *p* = 0.027) [77]. 

The discussed studies demonstrated how ML and DH can lead to improved therapeutic management of IBD. ML algorithms were able to predict treatment response and personalize medication dosing [90,91], whereas DH applications can improve treatment adherence and QoL [76,77], and guide therapies remotely [75]. Treatment strategies enhanced by DH and ML can reduce costs and burden of care due to their potential for providing personalized care for each patient at a low cost.

### 3.4. Monitoring 

#### 3.4.1. AI in Monitoring

Longitudinal monitoring of IBD is an important contributor to helping a patient achieve remission and to detecting complications in a timely manner because of the potential for early recognition of a flare-up [104]. Patients who are monitored consistently or attend regular appointments frequently experience better disease activity outcomes than those with less frequent appointments [105]. The use of AI can improve monitoring of IBD patients in order to avoid disease exacerbation and deliver appropriate medical care. 

AI can incorporate medical data to better assess disease activity among patients with IBD. In a study of 187 UC patients, a proprietary ML algorithm, referred to as a computer aided diagnosis (CAD) tool, used endocytoscopy images as input to detect persistent histological inflammation and disease activity [92]. Histological inflammation is a risk factor for clinical exacerbation and colorectal dysplasia and can be used as a prognostic for UC outcomes. The CAD tool achieved an overall diagnostic sensitivity of 74% (95% CI, 65–81%), specificity of 97% (95% CI, 95–99%), and accuracy of 91% (95% CI, 88–93%) at detecting histologic inflammation. The ability to effectively predict persistent histologic inflammation demonstrated the diagnostic potential of the system, with the potential to reduce the burden on pathologists by automating the process of grading intestinal inflammation of UC patients [92].

Similarly, a deep neural network for evaluation of UC (DNUC) was designed to assess whether patients were in endoscopic or histological remission [89]. The DNUC was trained using 40,758 endoscopic images and 6885 biopsy images collected from 2012 UC patients. To test their approach, 4187 endoscopic images and 4104 biopsies from 875 patients were used. The DNUC predicted endoscopic remission with 90.1% accuracy compared to experts and histologic remission with 92.9% accuracy, without requiring a mucosal biopsy. The authors suggest that the application of the DNUC can mitigate unnecessary biopsies due to its capacity to accurately assess histologic remission, which would reduce healthcare utilization and associated costs [89].

#### 3.4.2. DH in Monitoring

To minimize the burden of and increase accessibility to regular follow-up appointments required to monitor IBD activity and progression, DH methods such as distance management, tele-rehabilitation, and web applications are being studied as promising alternatives to outpatient appointments. Telemonitoring technologies represent an important avenue of patient care that have been explored at length by several researchers. The wide variety of options that DH can offer to support IBD patients will be addressed in this section.

A few studies have assessed the use of telemedicine systems, such as myIBDcoach, to improve patient’s disease monitoring and reduce healthcare utilization. MyIBDcoach is an application that allows IBD patients to longitudinally record patient-reported outcome measures (PROs), such as disease activity. Its ability to restructure the collected information in an informative and accessible manner for both users and healthcare professionals allows for improved disease monitoring as compared to oral recounting during a patient consultation [49]. In addition, there are web-based learning modules and a messaging platform to communicate with healthcare providers in order to improve patient disease knowledge and help them feel empowered in their illness experience. A randomized controlled trial was conducted across four healthcare institutions in the Netherlands [49]. Of the 909 IBD patients included in the trial, participants were randomly assigned to myIBDcoach (*n* = 465) or standard care (*n* = 444). After 12 months, a mean of 1.26 outpatient appointments with gastroenterologists were recorded in the telemedicine group, which was significantly lower than the 1.98 recorded in the standard care group, corresponding to an intervention effect of −0.72 (95% CI, −0.87 to −0.56) (*p* < 0.0001). In addition, patients who followed myIBDcoach also had significantly fewer hospital admissions overall: 16 patients who utilized telemedicine and 29 standard care patients were admitted to hospitals during the 12 months of the trial (*p* = 0.046). Hospital admissions were due to disease exacerbation or complications such as intestinal obstruction, surgery, side effects of treatment, and abdominal pain in patients with inactive disease. Overall, myIBDcoach maintained quality of care and disease monitoring while reducing healthcare utilization [49]. Therefore, increased frequency of monitoring made possible by telehealth approaches contributes to the prevention of disease exacerbation.

A separate randomized controlled trial also demonstrated the impact of remote monitoring using telemedicine on patients with IBD [48]. A web-based system, Telemonitoring of Crohn’s Disease and Ulcerative Colitis (TECCU), nurse-assisted telephone care, and standard care with in-person appointments were employed to monitor 21 patients each. The TECCU platform required patients to answer questionnaires about their symptoms and to report changes since their last evaluation, from which specialized healthcare professionals would create personalized recommendations for each patient to modify their therapies if necessary [48]. These questionnaires were completed after visits at baseline and at 12 and 24 weeks of the study. The primary outcome measured in this study was remission after 24 weeks of the intervention. This was measured with a modified Harvey–Bradshaw Index (HBI) for CD patients and the Simple Clinical Colitis Activity Index (SCCAI) for UC patients during remote checkups and the partial Mayo score for in-person visits during the study period. The investigators also measured CRP and FC levels and conducted complete blood analyses with nutritional profiling. After 24 weeks, when referring to HBI and Mayo scores (odds ratio = 0.12, 95% CI = 0.003–2.162, *p* = 0.19) and HBI and SCCAI scores (odds ratio = 0.11, 95% CI = −0.004 to 1.55, *p* = 0.13), there was better improvement in disease activity among patients using TECCU than those receiving standard care. Overall, 81% of patients in the TECCU group were in clinical remission, whereas only 66.7% and 71.4% of patients were in remission in the telephone and standard care groups, respectively. The reduction in disease activity was also associated with a greater decline in FC levels among patients using TECCU compared to standard care patients (estimated intervention effect: odds ratio = −0.90, 95% CI = −1.96 to 0.16, *p* = 0.11). Moreover, the amount of outpatient clinic visits was lower: patients in the TECCU intervention group accrued a total of 72 visits, whereas patients in the telephone care group had 85 visits and in the standard care group 131 visits. Therefore, TECCU demonstrated its potential to monitor the health of patients with IBD in order to improve their outcomes and minimize healthcare utilization [48].

Similarly, Marin-Jiménez et al. (2016) evaluated the diagnostic performance of self-reporting SCCAI to enable UC patients to self-assess and monitor their disease through a web platform named CRONICA. A total of 199 UC patients with evidence of disease activity that necessitated hospitalization or corticosteroid use in the 12 months prior to the study were selected to participate. This entailed completing online SCCAI evaluations individually as well as with a gastroenterologist. The online SCCAI questionnaires completed through CRONICA proved to be a reliable metric of UC disease activity: there was a good correlation between SCCAI scores self-reported by patients and those generated during a consultation with their physicians, as these results represented 85% agreement between assessments of remission or disease activity (95% CI: 80.8–88.6). Self-administered SCCAI via an online platform is another potential option for remote monitoring that can benefit UC patients [78].

In the same vein, Li et al. (2017) evaluated whether an IBD telemedicine clinic would provide a high-quality, low-cost alternative to traditional care. The telemedicine clinic was characterized by a virtual appointment with an IBD specialist through any device connected to the internet. Patients completed a pre-visit survey about their disease activity and a post-visit survey about their experience during the appointment, the time and money conserved by not having to commute to the visit, and preferences for future visits. The 48 participants in the study were IBD patients with a mean disease duration of 12.7 years, including 34 CD patients and 14 UC patients. Of the patients, 81% lived over 25 miles from their clinic and reported an average of $62 lost due to travel for in-person appointments. Although there were no significant differences in steroid use or biological therapies before and after telemedicine, 77% of participants preferred follow-up telemedicine appointments and continued to utilize it after the study. This demonstrated the feasibility of remote monitoring as well as its potential to reduce work or school absenteeism and financial burden, as patients saved an average of a half or full day of time compared to face-to-face visits, without sacrificing quality of care [79].

Another study with 59 participants found that DH platforms help patients gain an understanding of their disease and reduce healthcare utilization compared to standard care [80]. In the health monitoring application HealthPROMISE, patients could record and monitor their symptoms, medications, QoL scores measured by the Short Inflammatory Bowel Disease Questionnaire (SIBDQ), emergency room visits, and hospitalizations. After one year of data acquisition, there was no statistically significant difference between baseline and post-study SIBDQ scores, indicating that QoL did not improve with the use of the HealthPROMISE application relative to standard care. A potential limitation, however, is that only 32 out of 59 participants, of which 23 were CD patients and 9 were UC patients, logged into HealthPROMISE at least once, while the other 27 did not use the application. However, patients expressed significantly better understanding of the nature and course of their health condition (*p* = 0.026) and there was a statistically significant decrease in IBD-related hospitalizations and emergency room visits. At baseline, 25% of participants received hospital care, whereas after 12 months, there was only one visit recorded (3% of participants) (*p* = 0.03). This decrease in visits can be explained by the fact that the scores from the questionnaires incited the medical team to anticipate a decline in patients’ health earlier, preventing visits to the hospital, demonstrating the preventative potential of DH approaches [80].

Cross et al. (2019) had similar results as the previous study when assessing the impact of TELEmedecine for Patients with Inflammatory Bowel Disease (TELE-IBD). There were 348 participants enrolled in this randomized controlled trial, including 117 people in the control group, 115 people receiving text messages from TELE-IBD every other week, and 116 receiving these messages weekly. Participants in the TELE-IBD groups received educational tips about IBD as well as periodic general health messages twice monthly. UC or CD patients with worsening symptoms in the 2 years preceding the study were eligible. Those who received TELE-IBD messages experienced a decline in IBD-related hospitalizations and clinic visits, whereas patients in the control group had more hospitalizations over the course of a year. There were significantly fewer hospitalizations among TELE-IBD patients who received weekly messages compared to standard care. All CD patients, irrespective of the intervention they received, experienced improvements in their disease activity, whereas in the UC group only those who did not receive TELE-IBD support had reduced disease activity. Furthermore, all participants experienced a rise in QoL, yet only those who received TELE-IBD every other week had a significantly better QoL, as measured by the IBD Questionnaire (*p* = 0.03). The authors suggested that TELE-IBD did not significantly improve disease activity and QoL overall because more than half of the patients had inactive disease, which reduces the value of telemedicine as complementary care for these individuals. The need for telehealth and frequent monitoring increases with patient disease activity, which has a demonstrated effect on QoL of patients. [81].

Meanwhile, Quinn et al. (2019) qualitatively evaluated TELE-IBD through the lens of participants’ perceptions of the approach. A total of 348 IBD patients whose symptoms had worsened in the two years prior to the study received TELE-IBD messages weekly, every other week, or not at all, instead receiving standard care. Discussions with the participants following the use of this intervention revealed that patients saw benefits of this DH system, which included the opportunity to gain an enhanced understanding of IBD, to better monitor their symptoms, and to feel connected to their clinician. The patients confirmed that the TELE-IBD system is a useful component in IBD self-management and a helpful supplement to traditional follow-up appointments [82].

Similarly, the DH intervention IBD-Home was tested for its feasibility in clinical practice. IBD-Home consists of a mobile application and home-based FC test kit to help patients monitor their disease by reporting symptoms and receive assessments of their intestinal inflammation [83]. The randomized controlled trial measured patient satisfaction and adherence to the application; however a low compliance rate of 29% among the 84 IBD patients who received IBD-Home paired with standard care was reported. In fact, 43% of women in the study were compliant to IBD-Home, yet only 17% of men were compliant (*p* < 0.001). ‘Compliers’ were characterized as patients who completed at least one Swedish Inflammatory Bowel Disease Register (SWIBREG) symptom questionnaire and one FC test. The investigators suggested that this lack of compliance was likely due to the fact that the patients were in long-term remission, therefore they found IBD-Home redundant while concurrently receiving standard care. However, 33% of patients who used IBD-Home had increased their medical treatment, which is significantly higher than the 17% patients reported in the control group (*p* = 0.007). In addition, after controlling for age, gender, and FC, the intervention group had higher medical treatment than the control. An increase in treatment was associated with compliance, suggesting that monitoring of compilers’ disease activity was effective [83].

Although Östlund et al. (2021) identified a lack of willingness to use DH in patients in remission, follow-up care is an important aspect of long-term IBD monitoring due to its potential for prevention of future disease relapses. McCombie et al. (2020) demonstrated the usefulness of DH applications to mitigate the lack of follow-up monitoring for IBD patients. Their randomized controlled trial highlighted the ease of use for patients and physicians of several smartphone applications, including two symptom self-monitoring interfaces: IBDSmart and IBDoc. IBDSmart is used to evaluate disease activity through metrics such as the Harvey–Bradshaw Index and the Simple Clinical Colitis Activity Index, along with providing educational materials. IBDoc is an at-home tool to measure FC levels from stool samples. A total of 100 patients met the criteria for the study: a confirmed diagnosis of CD or UC and had at least two outpatient appointments and fewer than three disease flares in the past year. Sixty-three participants were allocated to the DH group and 54 were assigned to receive standard care. Those using DH applications had a significantly reduced amount of outpatient appointments after one year, with a mean of 0.6 appointments to gastroenterologist among DH users compared to 1.7 among control patients and did not cause a worsening of health outcomes or health-related quality of life (*p* < 0.001). Perfect adherence, which refers to the completion of all questionnaires provided in the applications, was recorded among users of both applications. A 50% perfect adherence to IBDSmart, 30% perfect adherence was associated with the use of IBDoc. Lack of personal appeal to remote monitoring and problems with app functionality were among the reasons that patients did not substantially adhere to IBDDoc. Overall, the investigators demonstrated the feasibility of these DH applications to monitor patients remotely and thus improve access to healthcare and reduce the need for in-person outpatient appointments [84].

The collection of studies discussed highlight the benefits of monitoring IBD with AI and DH tools. The use of AI techniques, such as a CAD tool and DNUC, can provide accurate evaluations of endoscopic images and biopsy results to predict endoscopic and histologic remission, enabling monitoring of disease activity without adding to the burden of care [89,92]. Telemedicine technologies that promote web-based and distance management of IBD enable longitudinal disease monitoring, which has improved the QoL and remission rates of patients while reducing the need for clinic visits and hospitalization [49]. Furthermore, through telemanagement and -monitoring, a variety of positive outcomes have been relayed, such as improved disease activity [48], prevention of hospitalizations [80,81], and monetary savings from virtual appointments [79]. Although not all DH technologies provide equal benefit or improvements compared to standard care, the quality of care that these tools can provide demonstrates their viability as a strong alternative to traditional clinic visits, reducing healthcare utilization and loss of productivity, among several other advantages to patients and healthcare systems.

### 3.5. Prognosis

Similar to many other chronic illnesses, patients with IBD are susceptible to disease-related complications and variability in disease progression [45,104]. The variability in disease progression can reduce a patient’s QoL as they may need to significantly change their lifestyle if entering a flare-up. Therefore, predicting disease activity provides valuable insight and enables patients and their medical team to make more informed decisions about their care and maintain QoL. However, due to the multivariate, complex nature of IBD, each patient’s experience is slightly different and it can be difficult to identify potential triggers with conventional, standard of care. ML methods and DH technology can be utilized to evaluate prognosis and thus impact treatment approaches.

#### 3.5.1. AI and Prognosis

There are several ML approaches that have demonstrated value for the prognosis of IBD patients. Firstly, an RF model was constructed to anticipate IBD-related hospitalization and corticosteroid use from longitudinal electronic medical records of 20,368 patients diagnosed with IBD [93]. This model used IBD-related patient data such as age and longitudinal laboratory data, including but not limited to serum albumin, CRP and CBC levels in order to predict the amount of IBD-related hospitalizations and number of outpatient corticosteroid prescriptions. The RF longitudinal model was a strong predictor of IBD-related hospitalizations and outpatient steroid use, with an AuROC of 0.85 (95% CI, 0.84–0.85). Therefore, this model has potential to assess patients’ risk of disease flare-ups and thus personalize treatment approaches [93].

#### 3.5.2. DH and Prognosis

Another study aimed to predict complex CD outcomes such as hospitalizations, unscheduled hospital visits, and bowel surgery, reported to a mobile monitoring system based on disease activity patterns reported to the same system. The prospective observational study had 266 CD patients record their symptoms over the course of 44 months through their smartphones in a web-based symptom diary for CD (CDSD) [85]. Through the analysis of these symptoms, they were able to conclude certain disease activity patterns, such as an increase in or persistently high activity of disease after 44 months were associated with worse clinical outcomes. Patients who reported increased disease activity throughout the study experienced significantly more hospitalizations (*p* = 0.004), unscheduled visits (*p* = 0.005), and bowel resection surgeries (*p* < 0.001) compared to the 220 CD patients who reported milder disease activity patterns (decrease in or continuously low activity or in remission at the time of the follow-up). This study demonstrated that disease activity reported by patients in a CDSD platform can serve as a prognostic tool to predict their clinical outcomes, indicating the practicality of digital health monitoring [85].

Overall, ML methods and DH technologies both present opportunities for accurate prognosis of IBD patients. Effective assessments of patient prognosis promote improved and personalized monitoring and therapeutic management, emphasizing the value of emerging technologies to improve how patient health and IBD-related outcomes are predicted [93]. RF models and symptom diaries are examples of applications that serve this function. By utilizing data about medication use, frequency of hospitalizations, inflammation, and serum albumin levels, both AI and DH can continue to become more adept at predicting patient outcomes [85,93].

## 4. Discussion

### 4.1. Summary of Evidence

The complex factors that contribute to a patient’s experience with IBD highlight the challenges presented to healthcare professionals as well as patients throughout the processes of diagnosis, treatment selection, monitoring, and prognosis of each case of IBD. The variables involved in the disease activity of and care for IBD patients, such as genetics, the microbiome, the environment, medication usage, and socioeconomic and psychological factors, demand different types of interventions in order to support all patient subtypes [106]. To this end, the studies in this review demonstrated that DH and AI provide several advantages for IBD healthcare. The opportunities for remote disease management, self-monitoring, treatment adherence, and reduced burden of care are expanding and show great promise. By providing patients with more efficient or accessible care, healthcare utilization can be optimized without sacrificing quality of care. Patients using DH platforms have demonstrated an improved medication management, treatment adherence, and quality of life [49,75,76,77,81]. Several studies also showed that DH methods can reduce the need for in-person appointments, decreasing the use of healthcare resources without compromising the standard of care [49,84]. DH technology such as applications and virtual clinics also provide timely access to healthcare resources and guidance [77,79].

However, some DH interventions did not have a significant impact on certain groups of patients, with limited effect primarily with those in remission. For instance, the IBD-Home application and remote FC test kit had a low compliance rate, which investigators attributed to users being in long-term remission. This can be justified by the notion that patients in remission may not be motivated to use DH tools because their disease activity has been managed well [83]. Adherence to other DH services such as IBDoc was also low due to users not being suited to remote monitoring and problems with the functionality of the app. However, IBDoc reduced healthcare utilization without negatively impacting the health and quality of life of patients, indicating that it provides value as a DH tool for those who utilize it [84].

The critical advantages that DH offers compared to standard care are the opportunities for remote care, accessibility to health services, monitoring, and prevention of complications. DH can reduce obstacles that chronic disease patients face in receiving longitudinal care, such as lack of access in rural and remote regions and the financial burden of disease. As a result, DH limits absenteeism, travel costs, and healthcare utilization, which further benefits society by conserving healthcare and financial resources [54,107].

Meanwhile, the benefits of DH are also reflected in the applications of AI for IBD. RF models, neural networks, and matrix factorization are among the many AI methods that can be applied to IBD data modalities. One valuable way that this has benefited the care of IBD patients is through the analysis of endoscopy imaging. AI can facilitate the differentiation between CD and UC and the assessment of the severity of IBD with high accuracy, through the determination of histological inflammation from biopsies [88,92]. AI-enabled approaches represent opportunities to make more objective assessments of mucosal inflammation and damage, and more accurately evaluate the presence of an IBD. Generally, endoscopic assessments vary between specialists due to biases and variable experience, which can lead to inconsistent findings. Therefore, AI systems, such as the deep neural networks developed by Takenaka et al., have the potential to help clinicians diagnose IBD, and further serve to train the gastroenterologists to evaluate endoscopic activity in a more consistent manner [89]. In order to achieve this potential, it is important to reduce biases in and use a large, heterogeneous dataset to train AI algorithms prior to clinical implementation. Moreover, RF models have also been employed to guide the treatment process by predicting the efficacy of certain medications [90,91]. This can reduce the work burden on pathologists and gastroenterologists, as well as the financial burden to healthcare systems, and improve the efficiency with which the optimal treatment plan for each patient is selected. Furthermore, research on the use of genetic and microbiome datasets to develop AI models continues to emerge. In this review, an example of the implementation of this data was the integration of exome sequencing data into a predictive model to distinguish healthy individuals from individuals with CD [86]. Biological annotation of genetic data can improve knowledge of IBD and lead to more personalized care for patients.

Overall, AI can enhance and optimize the diagnosis process and enable better characterizations of prognosis and treatment response. These assessments coincide with the stable monitoring offered by DH technologies, which can provide guidance on medication use and help predict patient prognosis. Further investigations on the impact of genetic, microbiota, endoscopic, and biomarker data will continue to refine approaches to AI and DH, thus improving the quality of care that IBD patients receive.

### 4.2. Limitations of Current AI and DH

There are several limitations to implementing DH and AI in the treatment of IBD. To begin with, more studies need to be performed to validate DH, whether in the form of mobile application or a virtual clinic, to evaluate their potential as an alternative treatment strategy for IBD [108]. Another limitation is that presently, it is difficult to rely on the results provided by the AI algorithm, due to insufficient validation in large and heterogeneous cohorts. In addition, most AI algorithms currently suffer from lack of interpretability, which further complicates their integration into the clinic [109]. Indeed, there needs to be further research on the competency, feasibility and interpretability of AI-assisted decision making [59].

A common limitation of AI algorithms is their large data requirement for achieving a high precision. Indeed, a considerable amount of data points with accurate labels provided by specialists is required to train a clinical grade algorithm, which is rarely the case. Therefore, an important step to establishing better AI algorithms that can help in diagnosing the IBD conditions in any clinical setting is the standardization of diagnostic definitions and producing large datasets with labels in accordance with these definitions [110]. Similarly, it is difficult to obtain data to develop predictive tools due to the prospective nature of prognostic studies. Factors that contribute to this issue include patient reluctance and lack of adherence to sharing self-reported symptom data and non-unified data, requiring manual processing that leads to statistical noise. Such noisy data can be challenging to transform into a mathematical format that can be processed by AI. Therefore, constructing accurate prognoses for IBD patients can be complicated. A potential solution to overcome this difficulty would be to ensure the alliance of clinicians, statisticians, and bioinformaticians to generate study protocols and create standardized datasets that are applicable to AI algorithms [59].

Another issue in prognostic studies that was not addressed in the articles reviewed but is significant is the possibility of missing data. Because prognostic studies by nature are long-term and involve a level of commitment (regular in person testing, completing questionnaires) there can be a drop in sample size. The loss of study participants can weaken the significance of the results. In addition, unjustified dropouts, biases, and missing information can further diminish the power of the results [111]. A potential solution for this issue would be to implement frameworks that are based on theoretical models of prognostic pathways and follow specific guidelines for prognostic studies. The frameworks and guidelines will be better suited for the type of research that is being performed with more favourable outcomes [112,113]. However, these frameworks and guidelines are beyond the scope of this paper.

There are also limitations for monitoring IBD through DH. E-health interventions such as telehealth and web-based applications have not consistently proven their efficacy in clinical, real-world settings. It has been suggested that e-health interventions have been designed without a functional framework for translation into care beyond the scope of a research project. Additional research is required in order to elucidate which factors lead to success of an e-health intervention and which aspects of research methods, such as assessing external validity of results and reporting details of the intervention studied, can be implemented more effectively to improve real-world practicality of DH [114].

## 5. Conclusions

The articles in our review demonstrate preliminary evidence of the potential of DH tools and AI methods in the context of IBD. Telemedicine options, such as virtual clinics, web-based services, and mobile applications provide patients with the ability to self-monitor their illness and receive care from specialized healthcare professionals at a distance. Endoscopic and histologic imaging, exome sequencing, and laboratory biomarker data have been utilized to create AI approaches to effectively diagnose patients as well as predict treatment response and disease activity. Therefore, DH and AI both offer advantages to the quality of care that IBD patients currently receive, leading to improved quality of life, less healthcare utilization, and better IBD-related health outcomes. However, the majority of these studies were performed in a regulated clinical environment. Therefore, further validation of these results in a more practical environment and at a larger scale is required in order to assess their efficacy in the general IBD population.

## Figures and Tables

**Figure 1 genes-12-01465-f001:**
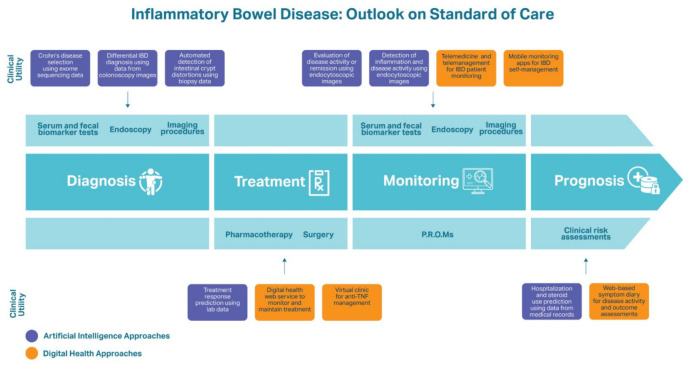
An overview of the traditional approaches to care for IBD patients and the novel digital health and artificial intelligence approaches reviewed.

**Figure 2 genes-12-01465-f002:**
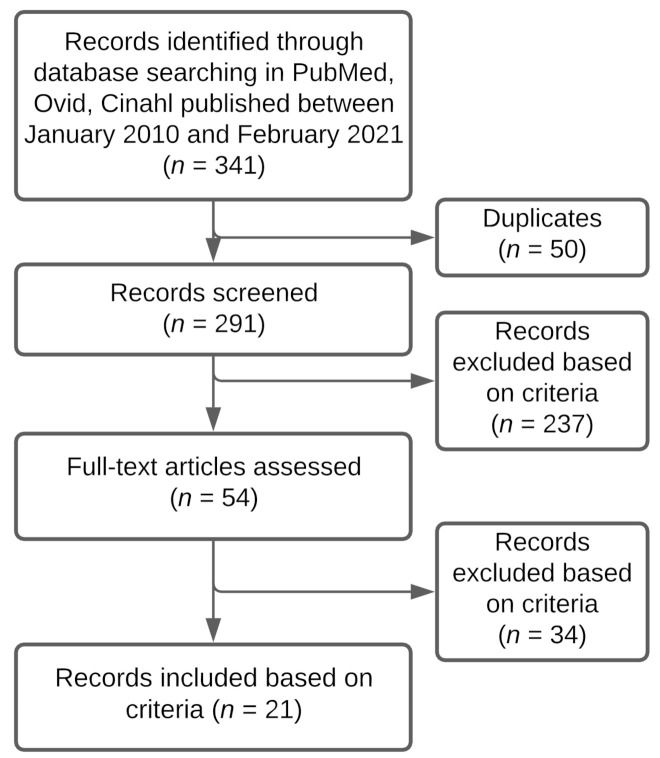
Selection process for articles about IBD and DH or AI in diagnosis, treatment, monitoring, or prognosis. Describes the trajectory of reviewing articles based on screening process and selection criteria.

**Table 1 genes-12-01465-t001:** Summary of key findings from reviewed articles. Characterizes each article in the review by their focus on digital health (DH) or artificial intelligence (AI) as a discipline, the aspect of inflammatory bowel disease (IBD) care that it addresses (diagnosis, treatment, monitoring, prognosis), and the key finding(s). Underlined and italicized are the categories of each approach to IBD care. In bold are the specific approaches utilized by the investigators. Abbreviations: Crohn’s disease (CD), area under the curve (AUC), random forest (RF), ulcerative colitis (UC), artificial intelligence (AI), inflammatory bowel disease (IBD), area under the receiver operator curve (AuROC), tumor necrosis factor (TNF), machine learning (ML), Telemonitoring of Crohn’s Disease and Ulcerative Colitis (TECCU), Simple Clinical Colitis Activity Index (SCCAI), TELEmedecine for Patients with Inflammatory Bowel Disease (TELE-IBD), quality of life (QoL), fecal calprotectin (FC), and health-related quality of life (HRQoL).

	Digital Health	
Diagnosis		
Treatment	Treatment Adherence and Maintenance “Constant-care” web service Significantly improved adherence to 5-aminosalicylate treatment, knowledge of IBD, and QoL compared to patients receiving standard care [75]. Helped UC patients optimize their maintenance treatment using mesalazine and improve treatment adherence, disease activity, and QoL [76].	
	Treatment Management Virtual clinic for anti-TNF therapy management Significantly shortened time until treatment success, provided suitable dose intensification, improved disease control, and improved treatment de-escalation compared to standard CD care [77].	
Monitoring	Telemedicine and Telemanagement Approaches	Mobile Applications
	myIBDcoach Significantly reduced the number of outpatient visits compared to IBD patients using standard care while maintaining QoC and disease monitoring (*p* < 0.0001) [49]. TECCU Reduced outpatient clinic visits among IBD patients. TECCU users experienced improvements in disease activity and 81% of these patients were in clinical remission by the end of the study, compared to 71.4% of patients receiving standard care [48]. CRONICA The self-administered SCCAI via the CRONICA web platform was a trustworthy self-assessment tool for UC patients to monitor their. Online SCCAI scores were 85% in agreement with physician’s assessments of remission or UC disease activity [78]. IBD telemedicine clinic Appointments were evaluated to assess the quality of care provided at a low cost in comparison to standard care. Telemedicine patients saved a mean of $62 in travel costs and at least half a day of time without negative impacts on quality of care [79].	HealthPROMISE Led to a significant reduction in hospitalizations and emergency room visits within one year among IBD patients compared to those who received standard care [80]. TELE-IBD TELE-IBD groups experienced a decline in IBD-related hospitalizations, with a significant decrease when receiving TELE-IBD messages weekly compared to standard care. TELE-IBD educational messages did not significantly improve disease activity and QoL in comparison to standard care, potentially due to the patients having more severe CD and UC [81]. Interviews with patients using TELE-IBD revealed that they considered the service a beneficial supplement to traditional follow-ups and a useful component in IBD self-management to stay educated on IBD, monitor their symptoms, and connect with their physician [82]. IBD-Home 29% of patients were compliant to the IBD-Home application and FC test kit after one year. Patients who were compliant experienced a rise in medical treatment, providing the value to remote disease monitoring [83]. Self-monitoring applications (IBDsmart and IBDoc) Led to significantly fewer outpatient appointments than standard care patients (mean of 0.6 vs. 1.7) without affecting health outcomes or HRQoL (*p* < 0.001) [84].
Prognosis	IBD-Related Predictions Web-based symptom diary for CD Patient-reported IBD-related symptoms were associated with significant increases in hospitalizations, unscheduled visits, and bowel resection surgeries among CD patients with more severe disease [85].	
	Artificial Intelligence	
Diagnosis	IBD Detection Tri-matrix factorization model used a combination of exome sequencing data and biological knowledge to differentiate healthy individuals from CD patients (AUC = 0.816) [86]. RF model differentially diagnosed CD and UC using descriptions of colonoscopy images (AUC = 0.936) [87]. AI system built using a probabilistic neural network assessed intestinal crypt architecture distortion and mucosal damage from patient biopsies and diagnosed IBD with 98.31% precision and recall [88]. Deep neural network for evaluation of UC predicted endoscopic remission with 90.1% accuracy and histologic remission with 92.9% accuracy using endoscopic images and biopsies from UC patients [89].	
Treatment	Treatment Response Predictions RF algorithms predicted clinical responders and non-responders (AuROC = 0.856) and non-adherence to thiopurine therapy (AuROC = 0.813). Can be used to personalize thiopurine dosages [90]. RF model predicted corticosteroid-free endoscopic remission at 52 weeks of vedolizumab treatment using data acquired during week 6 of therapy (AuROC = 0.73) [91].	
Monitoring	Inflammation and Disease Activity Monitoring Deep neural network for evaluation of UC predicted endoscopic remission with 90.1% accuracy and histologic remission with 92.9% accuracy using endoscopic images and biopsies from UC patients [89]. Proprietary ML algorithm was 91% accurate at detecting histologic inflammation from endocytoscopic images and therefore assessing disease activity and risk of clinical exacerbation [92].	
Prognosis	IBD Assessment and Predictions Proprietary ML algorithm was 91% accurate at detecting histologic inflammation from endocytoscopic images and therefore assessing disease activity and risk of clinical exacerbation [92]. RF model constructed from medical records of IBD patients predicted IBD-related hospitalizations and outpatient steroid use (AuROC = 0.85) [93].	

The underlined and italicized terms should ideally be grouped with the text underneath it.

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
