# Peer review of "Role of Digital Health and Artificial Intelligence in Inflammatory Bowel Disease: A Scoping Review"

_genes, 2021, doi:10.3390/genes12101465_

Round 1
Reviewer 1 Report
This is a well written comprehansive review which may be informative for researchers dealing with inflammatory bowel diseases. The study was well planned, performed and written. The references consist up-to-date publication. The review may be accepted in its current form.
Author Response
Thank you for your positive comments on the manuscript.
Reviewer 2 Report
It is a deep and detailed overview of the benefits of digital health and artificial intelligence in the care of patients with IBD.
I’d like to note however that there are several recent reviews are out there in this field.
This area of healthcare is currently rapidly evolving. I hope that we will have more use of artificial intelligence to better predict the responsiveness of Crohn’s patients to biologic treatment like Infliximab, and the more recent ones.
I’d like to see if authors could elaborate on this a bit more in a few sentences and with a few examples.
I also felt while reading this manuscript that it contained some fluff here and there. Nevertheless, it has a clear and meaningful message.
Thank you.
Author Response
Thank you for the thorough review and assessment of our manuscript.
We appreciate the feedback and have provided the follwoing responses:
I’d like to note however that there are several recent reviews are out there in this field.
- Although there are several recent reviews in the field that discuss the role of artificial intelligence or digital health for IBD, to our knowledge, there are none that have incorporated both digital health and AI into their review.
This area of healthcare is currently rapidly evolving. I hope that we will have more use of artificial intelligence to better predict the responsiveness of Crohn’s patients to biologic treatment like Infliximab, and the more recent ones.
I’d like to see if authors could elaborate on this a bit more in a few sentences and with a few examples.
- The value of artificial intelligence for monitoring and predicting response to biologic therapies is certainly important. To elaborate on this as suggested, we included a paragraph dedicated to this topic in the introduction in section 1.3.2 on the Role of Artificial Intelligence in the Care of Inflammatory Bowel Disease. These concepts return in the results, so having the additional context with some examples in the introduction is helpful.
I also felt while reading this manuscript that it contained some fluff here and there. Nevertheless, it has a clear and meaningful message.
- To address the fluff or wordiness of some passages, we did another read-through and attempted to revise phrasing where possible.
Once again, thank you for the comments to help improve the manuscript.